# ClusterRadar: An interactive web-tool for the multi-method exploration of spatial clusters over time

**Lee Mason**[1,2]*, **Blánaid Hicks**[2], **Jonas S. Almeida**[1]

**1** Division of Cancer Epidemiology and Genetics, National Institutes of Health, Rockville, Maryland, United States of America, **2** Center for Public Health, Queen's University Belfast, Belfast, United Kingdom

* masonlk@nih.gov

**Data availability statement:** The code, example data, and documentation can be found at https://github.com/episphere/ClusterRadar.

## Abstract

Spatial cluster analysis is crucial for understanding localized patterns in geospatial data, with wide-ranging applications for scientific discovery and decision-making. However, the dynamic nature of spatial clusters and the diverse range of clustering methods available can make analysis and interpretation challenging. We introduce ClusterRadar, a web-based tool designed to streamline this process by uniquely prioritizing longitudinal analysis and multi-method comparison of spatial clusters. It empowers users to easily perform clustering with multiple methods, directly compare results, and visualize spatiotemporal patterns through a novel design of linked interactive visualizations. ClusterRadar aims to maximize utility to a broad user base by supporting various geospatial formats and executing entirely within the browser to ensure data privacy. ClusterRadar is available at https://episphere.github.io/ClusterRadar.

## Introduction

Space plays a critical role in many real-world phenomena, where the proximity of entities in a system often effects the strength and nature of interactions [1–3]. This idea is succinctly captured in Tobler's first law of geography, which states that "everything is related to everything else, but near things are more related than distant things" [3,4]. The field of geospatial analysis offers a powerful set of methods for understanding these relationships [5,6]. Techniques like spatial clustering, machine learning, spatial statistics, and network analysis allow analysts to uncover informative relationships within geospatial data [7–9]. The growing availability of geospatial data, coupled with an increasingly interconnected world, mean that sophisticated geospatial techniques are increasingly important in driving scientific discovery and supporting informed decision making [6,8].

One valuable application of spatial analysis is the detection of spatial clusters: groupings of neighboring geospatial features which exhibit significant similarity across certain attributes [8,10]. For example, a spatial cluster might be a collection of neighboring counties with similar rates of breast cancer. Spatial cluster detection is a widely employed technique in geospatial analysis, crucial for advancing scientific understanding of complex systems and tackling practical challenges [8,10,11]. Spatial clustering has been extensively applied across

**Funding:** The authors received no specific funding for this work. Research reported in this publication was supported by the National Cancer Institute of the National Institutes of Health (CAS 10901).

**Competing interests:** The authors have declared that no competing interests exist.

disciplines, including cancer cluster detection in epidemiology [12], crime cluster analysis in criminology [13], and the study of weather events in climatology [14]. Robust spatial cluster detection methodologies, particularly those founded on statistical rigor, empower analysts to expand beyond simple visual pattern recognition in maps [15].

The field of spatial cluster analysis continues to evolve, addressing the expanding analytical requirements of researchers across an increasingly diverse set of disciplines [5,6,16]. There is a growing need to expand the scope of spatial clustering methods, including the deeper integration of temporal data, and better support for the comparison of results across multiple methods. References [6,8,17,18]. This growing gap between the expanding needs of researchers and the limitations of existing tools highlights the need for novel solutions that can unlock the full potential of spatial cluster analysis. Tools that prioritize accessibility, flexibility, and the integration of temporal data are essential to meet the rising demand and foster new discoveries in this rapidly evolving field. References [16,19].

In this paper, we introduce ClusterRadar, a web-tool designed to meet the growing need for a user-friendly environment in which to analyze spatial clusters over time and across multiple methods. ClusterRadar runs fully on the client-side in the user's browser, avoiding the need for installation and preserving the privacy of the user's data. The core feature of the tool is an interactive dashboard consisting of five panels, each providing a different perspective on the methodological and temporal aspects of multi-method, spatiotemporal clustering results. This paper details the methodology, design, and implementation of the tool, emphasizing its unique focus on multi-method and longitudinal comparisons. By incorporating interactive visuals, ClusterRadar makes complex spatial clustering analysis accessible to a broader audience. ClusterRadar is available at https://episphere.github.io/ClusterRadar. To provide context for ClusterRadar's capabilities, the following section reviews the fundamental concepts of geospatial data, spatial cluster analysis, spatial autocorrelation, and local indicators of spatial association (LISAs).

## Background

**Geospatial data.** Geospatial data comprises data points explicitly linked to geographic locations on the Earth's surface. These references are typically given as points (representing precise coordinates), lines (representing paths or routes), or areas (polygonal regions) [20]. Geospatial data can employ various coordinate systems, such as latitude and longitude, to define spatial locations. Areal data, such as countries or census regions, is highly prevalent because the often standardized and familiar area definitions facilitate integration across multiple data sources [21]. While less common, line data (e.g., roads) sometimes holds analytical value. Point data can often be aggregated into areas, which can simplify analysis, improve statistical power, or facilitate linkage with other datasets [20].

**Spatial cluster analysis.** Spatial clusters are defined in a number of ways, depending on the type of geospatial data and the specific analytical goals being tackled. Broadly, a spatial cluster is a grouping of geographically related features that exhibit a substantial degree of concentration or similarity [8,10]. There are many different approaches to spatial clustering, including partition clustering, hierarchical clustering, density-based clustering, and LISA-based clustering [8,10]. LISA-based clustering methods use local indicators of spatial association (LISAs) to determine whether each location belongs to a spatial cluster [22,23]. They are popular due to their robust statistical foundation, interpretability, and wide-spread implementation in different analytical environments [8,24].

**Spatial autocorrelation.** Spatial autocorrelation is a fundamental concept in spatial analysis describing the extent to which geographically proximate locations exhibit similar attributes [23,25]. Spatial autocorrelation can be quantified using several different statistics,

the most prominent being Moran's I and Geary's C [26,27]. Both of these statistics indicate whether or not a dataset exhibits a greater degree of spatial autocorrelation than would be expected by random chance, but they take different numerical approaches to do so. Spatial autocorrelation can be positive, meaning that locations which are near each other tend to exhibit similar attribute values, or negative, meaning that locations which are near each other tend to exhibit dissimilar attribute values.

Global spatial autocorrelation statistics are only capable of indicating whether or not spatial autocorrelation is present in a dataset but not at which locations it occurs [23]. In reality, geospatial datasets often exhibit pockets of both positive and negative spatial autocorrelation, alongside regions which don't show any substantial spatial autocorrelation, and it can be useful to analyze this heterogeneity. For this purpose, local spatial autocorrelation statistics are required. Unlike global spatial autocorrelation statistics, local spatial autocorrelation statistics are a property of each location in the dataset, indicating whether a specific location is similar to its neighbors (a spatial cluster), or dissimilar to its neighbors (a spatial outlier). The majority of global spatial autocorrelation statistics have local counterparts, such as Local Moran's I and Local Geary's C [23]. Local spatial autocorrelation statistics are one (major) subset of local indicators of spatial association (LISA), which cover a broader range of local association relationships [23].

**Local indicators of spatial association.** A local indicator of spatial association (LISA) is a statistical measure that quantifies some degree of spatial association between a particular location and its neighbors within a geospatial dataset [23]. In addition to local spatial autocorrelation statistics, a prominent example of a LISA is the Getis-Ord Gi/Gi* family of statistics, local indicators which detect whether or not a location belongs to "hot-spot" or "cold-spot". Hot-spots are statistically significant clusters of high values, while cold-spots are statistically significant clusters of low values [28,29]. Unlike Getis-Ord Gi/Gi*, Local Moran's I and Local Geary's C are incapable of directly detecting hot-spots and cold-spots, they can only detect whether or not a location is significantly similar or dissimilar to its neighbors. On the other hand, Getis-Ord Gi/Gi* are incapable of detecting spatial outliers. While Local Moran's I and Local Geary's C are not directly capable of detecting hot-spots and cold-spots, additional analysis is often performed to distinguish hot-spots and cold-spots among areas exhibiting significant local spatial autocorrelation [23]. All of the aforementioned LISAs (Local Moran's I, Local Geary's C, and Getis-Ord Gi/Gi*) are frequently used for detecting spatial clusters, though the spatial clusters should be interpreted differently due to the different goals of the methods. For more information on the formulation and interpretation of these statistics, see Methods.

## Related work

**Temporal analysis of spatial clusters.** Spatiotemporal analysis of clusters offers a powerful approach to uncovering hidden patterns and dynamics within complex datasets, offering insights beyond what purely spatial or temporal approaches can provide [19,30,31]. While specialized spatiotemporal methods exist for analyzing the evolution of spatial distributions [32–36], their complexity can limit accessibility and interpretability [33]. A more accessible approach involves applying established spatial analysis techniques longitudinally, comparing results across different time points [37–41]. However, this powerful form of comparative spatiotemporal analysis is currently hindered by a critical gap in software tools. Existing tools require analysts to manually manage and compare results across time—a time-consuming, error-prone process that limits the potential of this approach. This work directly addresses this gap, enabling more streamlined and robust spatiotemporal insights.

**Multi-method analysis of spatial clusters.** There are many different methods for detecting spatial clusters and they often produce vastly different results, making the choice of method a crucial and often challenging task in spatial analysis [8,17,17,18,18,23,29,42–44]. Due to differences in results from different LISA-based clustering methods, many comparative works recommend that multiple methods be applied in tandem [17,18,29,44,45]. This advice is extensively followed in the literature, with many geospatial clustering analyses applying more than one method [41,46–48]. However, despite the popularity and utility of a multi-method approach, existing software implementations do not directly support the automatic, streamlined comparison of different clustering methods. This forces analysts into the time-consuming process of separately executing and manually synthesizing results, hindering the wider adoption of multi-method analyses.

**Interactive spatial dashboards.** Spatial dashboards are a popular environment for analyzing spatial data, with a growing body of work exploring novel designs, specific spatial analytical techniques and datasets [49–52], and challenges like color use [53,54], temporal data integration [55–59], and interactivity [60–62]. The COVID-19 pandemic broadened their use and appeal [49,55,63]. There is increasing interest in integrating statistical results, especially for detecting spatial clusters [15,16,64]. While most dashboards focus on single methods or datasets, this work addresses the need for comparative analysis of multiple clustering methods and their temporal evolution—a complex problem requiring specialized visualizations.

**Spatial clustering software.** Several applications and libraries support spatial cluster analysis. ArcGIS, a commercial software, offers support for Local Moran's I and Getis-Ord Gi* [65]. Open-source alternatives exist, such as QGIS [66] (with limited LISA-based clustering via plug-ins) and CrimeStat [67] (specializing in crime data analysis using methods like Local Moran's I and Getis-Ord Gi/Gi*). Programming libraries like PySAL (Python) and spdep (R) support LISA-based clustering but require coding expertise [68,69]. GeoDa, an open-source tool dedicated to spatial analysis, offers an interactive interface for the analysis and visualization of LISA-based clustering methods, but doesn't support temporal analysis or the direct comparison of different methods [70]. SaTScan supports spatiotemporal cluster analysis but focuses on scan statistic-based methods and doesn't support LISA-based approaches [71]. While most tools require local installation (some with limited support for different operating systems), web-based options like GeoDa-Web and its associated JavaScript library (jsgeoda) are emerging, though currently with limited features [72]. None of these solutions provide direct, user-friendly support for the longitudinal analysis of spatial clusters or the ability to easily compare the results of different spatial clustering methods within a single, integrated environment

## Design considerations

We have formulated five key design considerations based primarily on important challenges and analytical ideals identified in the literature.

- **D1: Representation of temporal dynamics of spatial clusters.** Spatiotemporal analysis is rapidly gaining importance across diverse disciplines [6,36,73]. Historically, a major shortcoming in spatial analysis has been neglecting temporal dynamics [74]. However, the growing availability of user-friendly spatiotemporal software is steadily overcoming this limitation [71]. One valuable approach involves the longitudinal analysis of spatial clusters over time, which reveals how spatial clusters evolve [39,40,75]. Despite the utility of this technique, software with dedicated support remains scarce. This highlights the need for applications that prioritize the clear representation of temporal dynamics within spatial clusters.

- **D2: Comparison of results from multiple spatial clustering methods.** Spatial clustering encapsulates a wide variety of approaches, often producing vastly different results [8,45]. There is limited theoretical guidance on how to select an appropriate method [8]. Comparative studies of spatial clustering methods often recommend using multiple methods simultaneously and interpreting the results in conjunction [17,18,29,44,45]—a common approach across multiple disciplines. This highlights the need for applications which perform multiple spatial clustering methods at once and facilitate the comparison of results across methods.

- **D3: Application of varied interactive graphical elements to simplify the analysis of complex results.** Representing temporal and multivariate geospatial results can be challenging to the competing demands of the various visual elements—this is a well-recognized problem in geospatial visualization [57,59,76,77]. A multi-plot, interactive dashboard can tackle these challenges by representing different aspects of the data in different visualizations, drawing on the unique strengths of each to improve overall clarity [77–79]. Interactivity further eases the complexity of this analysis, revealing details only as they are required, and helping users stay oriented while exploring the data [78,80]. This highlights the need for applications which employ multi-faceted, interactive graphics to facilitate the exploration of complex geospatial results.

- **D4: Goal focused and appropriately scoped design to ensure usability for non-expert users.** Powerful geospatial software often presents a steep learning curve for non-experts because it requires a robust understanding of complex geospatial concepts [16,81]. As geospatial methods becomes increasingly integrated into diverse analytical pipelines, the need for user-friendly geospatial analytical applications grows. To make geospatial methods more accessible, a goal-driven approach is required [81]. This approach focuses on the desired analytical outcomes, rather than requiring users to understand the low-level steps required to achieve them. While feature-rich geospatial software offers versatility, it can overwhelm non-experts with its complexity [82]. This highlights the need for simpler, goal-oriented tools designed for specific analytical tasks.

- **D5: In-browser web implementation to ensure FAIR distribution and privacy preservation.** There has been a recent emphasis in science on adherence to the FAIR principles (findability, accessiblity, interoperability, and reproducability), including for software [83, 84]. Due to its ubiquity, familiarity, and inherent support for information sharing, the web provides a naturally FAIR place in which to distribute software. However, server-reliant software requires the user to upload their data, which may violate privacy requirements of sensitive data (common in fields like epidemiology). This highlights the need for applications which are distributed on the web and run fully client-side inside the sandbox of the user's browser.

## Methods

### Preparation

**Normalization.** Given a list of spatially-referenced values $X = [x_1, x_2, ..., x_n]$ the first step is to z-score normalize each value, which simplifies the downstream calculations:

$$z_i = \frac{(x_i - \mu)}{\sigma} \tag{1}$$

Where $\mu$ and $\sigma$ are the mean and standard deviation over all values in the dataset.

**Weight matrix.** Local indicators of spatial association require a definition of the relationships between locations in the dataset. This is encapsulated in a weight matrix. Given a dataset with $n$ locations, a weight matrix is a $n \times n$ square matrix $W$ where element $W_{i,j}$ quantifies the relationship between location $i$ and location $j$. In its simplest form, a binary weight matrix takes $W_{ij} = 1$ if location $i$ is a neighbor of location $j$, and $W_{ij} = 0$ otherwise. The definition of a "neighbor" is the decision of the analyst: the most common approach uses simple areal contiguity. While other weighting schemes exist, the binary weight matrix acts as a simple and computationally efficient starting point for analysis, appropriate when a more nuanced weighting scheme between neighbors (e.g., inter-area traffic flow, shared border length) is not apparent or readily available. For simplicity in the later calculation, we will row-normalize the weight matrix:

$$\forall i \in \{1, 2, \dots, n\}, \quad \sum_{j=1}^{n} W_{ij} = 1 \tag{2}$$

Choosing an appropriate weight matrix is challenging and depends on the specific parameters and goals of the task at hand. Local indicators of spatial association are highly sensitive to the choice of weight matrix.

## Local indicators of spatial association

**Moran's I.** The Moran's I statistic is a popular local indicator of spatial association, implemented in most major geospatial software packages and libraries [24,26]. It measures spatial autocorrelation. Assuming a row-normalized weight matrix $W$, Moran's I is calculated as follows:

$$I = \frac{\sum_i^n \sum_j^n W_{ij} \cdot z_i \cdot z_j}{n - 1} \tag{3}$$

A significant negative value for the Moran's I statistic indicates negative spatial autocorrelation, a significant positive value indicating positive spatial autocorrelation, and a value close to 0 indicates no spatial autocorrelation (spatial randomness). The Local Moran's I statistic addresses the need for a more granular assessment of spatial autocorrelation by breaking the global spatial autocorrelation into a separate value for each location [23]. The Local Moran's I statistic for location $i$ is calculated as follows:

$$I_i = \frac{z_i \cdot lag_i}{n - 1} = \frac{z_i \cdot \sum_j^n W_{ij} \cdot z_j}{n - 1} \tag{4}$$

Note the spatial lag term, $lag_i = \sum_j^n W_{ij} \cdot z_j$. Spatial lag is a useful concept when interpreting local spatial autocorrelation; it is essentially the weighted mean of a location's neighbors.

A significant negative value for Local Moran's I indicates that the location is a spatial outlier (significantly different from its neighbors), a significant positive value indicates that the location belongs to a spatial cluster (significantly similar to its neighbors), and a non-significant value indicates that the location does not exhibit significant local spatial autocorrelation. Results can be further categorized by looking at how the value and spatial lag of a location compare to the mean, which is easy to do with z-score normalized values because the mean is equal to 0. The possible assignments are "high-high" (if the value and lag are both positive), "low-low" (if the value and lag are both negative), "high-low" (if the value is positive and the lag negative), and "low-high" (if the value is negative and the lag positive).

**Geary's C.** Like Moran's I, Geary's C is a statistic which measures spatial autocorrelation but the two methods differ in their approach: Moran's I measures spatial autocorrelation using the correlation between neighboring values whereas Geary's C measures spatial autocorrelation using the square differences between neighboring values [28,29]. Geary's C has less widespread support in geospatial software than Moran's I. Geary's C is calculated as follows:

$$C = \frac{\sum_i^n \sum_j^n W_{ij} \cdot z^2}{2n} \tag{5}$$

Geary's C takes values 0 or greater. The values are interpreted by their proximity to 1, with values less than 1 indicating positive spatial autocorrelation, and values greater than 1 indicating negative spatial autocorrelation. Values close to 1 indicate spatial randomness.

Like for Moran's I, there is a local equivalent of the Geary's C to provide a more granular assessment of spatial autocorrelation [85]. Local Geary's C is calculated as follows:

$$C_i = \sum_j^n W_{ij} \cdot (z_i - z_j)^2 \tag{6}$$

In essence, the Local Geary's statistic is a weighted sum of the squared difference between a location's value and its neighboring values. The statistic takes values 0 or greater. Unlike for the global Geary's C, the value of the local Geary's C statistic has no inherent meaning—the "no spatial autocorrelation" point is no longer equal to 1. Instead, it must be interpreted with significance testing: values that are significantly lower than expected indicate positive spatial autocorrelation, values significantly higher than expected indicate negative spatial autocorrelation, and values not significantly different than expected indicate spatial randomness. Like for Local Moran's, a Local Geary's C result can be further specified by inspecting the location's value and spatial lag. However, unlike for Local Moran's I this is not always possible. If the Local Geary's C statistic indicates positive spatial autocorrelation, then the following assignments can be made: "high-high" (if the value and lag are both positive), "low-low" (if the value and lag are both negative), and "other positive spatial autocorrelation" (the remaining cases). If the Local Geary's C statistic indicates "negative spatial autocorrelation" then this result can't be specified any further.

**Getis-Ord G.** The Getis-Ord G family of statistics differ from Moran's I and Geary's C in that they do not measure spatial autocorrelation, but instead directly measure hot-spots and cold-spots [29]. A hot-spot is a group of neighboring locations with significantly higher than expected values, whereas a cold spot is a group of neighboring locations with significantly lower than expected values. The Getis-Ord General G statistic is a global measure that indicates whether a geospatial dataset exhibits clustering overall and whether that clustering is generally of high values or low values. The Getis-Ord General G statistic is calculated as follows:

$$G = \frac{\sum_{i=1}^n \sum_{j=1}^n W_{ij} \cdot z_i \cdot z_j}{\sum_{i=1}^n \sum_{j=1}^n \cdot z_i \cdot z_j}, \quad \text{where } j \neq i \tag{7}$$

Getis-Ord General G must be interpreted in relation to a reference distribution (usually obtained using permutation testing, see Assessing significance). If the observed value of G is significantly higher than the expected value then the dataset exhibits overall clustering of high values (hot-spots), if the observed value of G is significantly lower than the expected value then the dataset exhibits overall clustering of low values (cold-spots)

The Getis-Ord Gi⋆ and Getis-Ord Gi statistics are two local statistics which indicate whether a specific location belongs to a hot-spot or cold-spot. The Getis-Ord Gi⋆ statistic is calculated as follows:

$$G_i^* = \frac{\sum_{j=1}^{n} W_{ij} \cdot z_j}{\sqrt{\frac{1}{n-1} \cdot \left[ n \cdot \left( \sum_{j=1}^{n} W_{ij}^2 \right) - 1 \right]}} \tag{8}$$

The Getis-Ord $G_i$ statistic is similar, except it doesn't include the value at the focal location:

$$G_i = \frac{\sum_{j=1}^{n} W_{ij} \cdot z_j - \bar{z}(i)}{S(i) \cdot \sqrt{\frac{1}{n-1} \cdot \left[ n \cdot \left( \sum_{j=1}^{n} W_{ij}^2 \right) - 1 \right]}} \tag{9}$$

Where $\bar{z}(i)$ and $S(i)$ are the mean and variance over all z-normalized values excluding the value at location $i$. The interpretation of Getis-Ord Gi and Gi⋆ is similar. If the observed value of G/G⋆ is significantly less than 0 then that location exhibits a clustering of high values (a hot-spot). If the observed value of G/G⋆ is significantly greater than 0 then that location exhibits a clustering of low values (a cold-spot). If the observed value is not significantly different from the expected value, this indicates that there is no spatial clustering of values.

### Assessing significance

In order to correctly interpret indicators of spatial association, it is necessary to determine whether or not a value is significant. The standard approach is to use a permutation test because, unlike an analytical derivation of the statistic's theoretical distribution, it avoids imposing unrealistic assumptions upon the data [23]. A permutation test on a spatial statistic involves shuffling the data values across the spatial locations a number of times, calculating the statistic for each shuffle, and then using those values to build an empirical distribution against which the actual value can be compared. When the actual value is compared to the permuted values, a pseudo p-value can be calculated. The interpretation of a pseudo p-value differs somewhat from the interpretation of a traditional, analytical p-value. It is important to remember that the value of $p^*$ depends on the number of permutations performed. The usual p-value cutoffs (e.g. $p<0.05$, $p<0.01$) are often used when interpreting pseudo p-values, but the analyst should be aware of the dependence on the number of permutations and avoid thinking of pseudo p-values as equivalent to traditional p-values. For this reason, we use $p^*$ to refer to pseudo p-values in this manuscript and within the ClusterRadar web-tool. For specific details on how pseudo p-values are calculated, see S2 Appendix (Assessing significance). In order to simplify the interpretation of the spatial statistics in the ClusterRadar tool, the values are z-score normalized with the permuted set of values $S$.

## Results

### ClusterRadar web-tool

ClusterRadar is a web-tool that allows users to perform spatial cluster analysis and inspect the results in an interactive dashboard. The tool is fully in-browser and executes entirely on the client side. ClusterRadar performs spatial clustering using several popular local indicators of spatial association: Local Moran's I, Local Geary's C, Getis-Ord Gi, and Getis-Ord Gi⋆. By default, the tool will perform only one of the Getis-Ord methods (Gi⋆) due to their similarity, but the user is free to enable and disable methods to suit their needs. ClusterRadar operates on areal geospatial data. The dashboard (see Fig. 1) consists of five plot panels, each providing

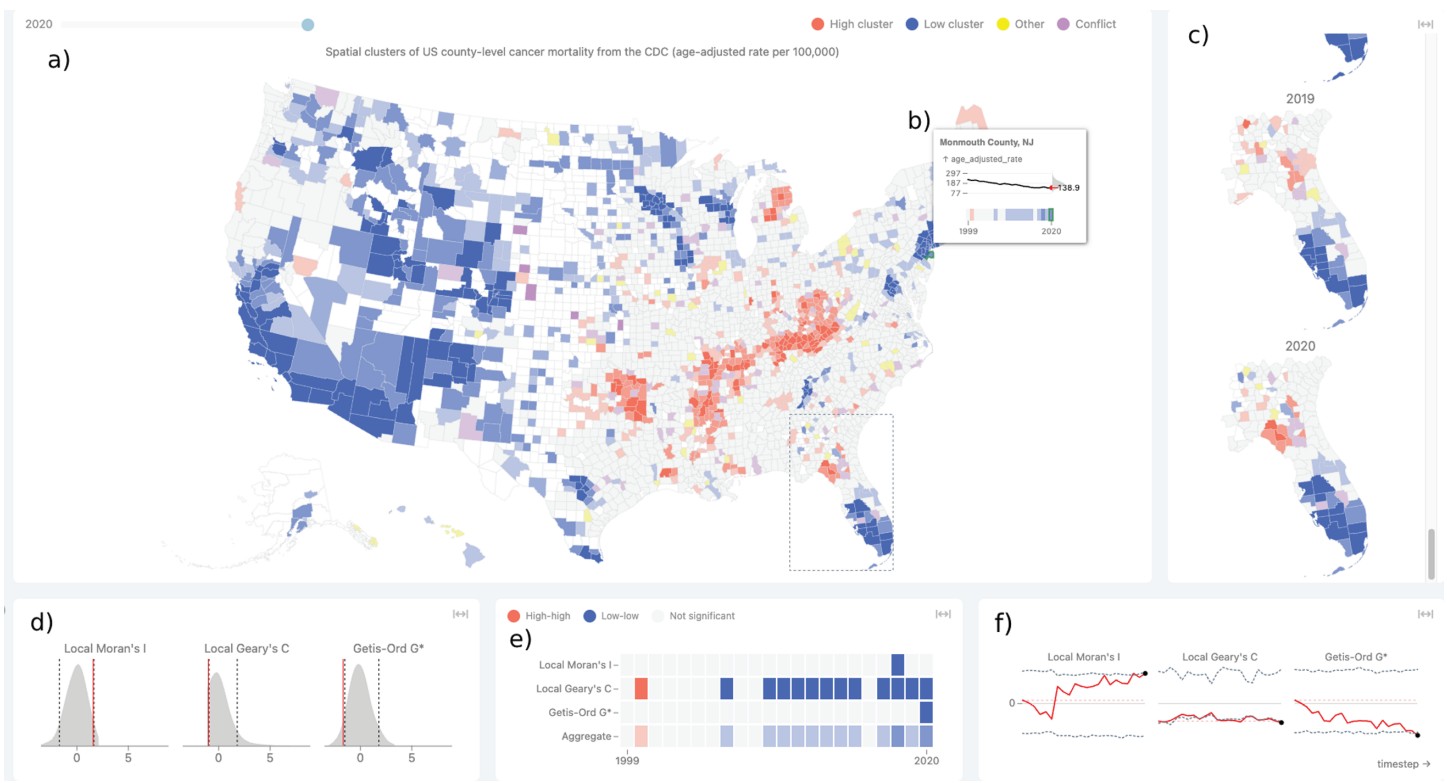

**Fig 1. The main dashboard of the ClusterRadar tool, showing the various plot panels and the graphical tooltip.** The dashboard is divided into two rows of plots: the panels in the top row show the geospatial distribution of the cluster assignments, and the panels in the bottom row show various additional details about the results. The five main panels are: (a) the primary cluster map panel, (c) the zoomed map reel panel, (d) the statistical density plot panel, (e) the cluster assignments cell plot panel, and (f) the statistical time-series plot panel. In this screenshot, the user has hovered over a location (Monmouth County, NJ) showing (b) the graphical tooltip, which gives additional information about that location's data and cluster assignments over time. Because the user is hovering over a specific location, the bottom row plots show results related to that location. The user has also selected a sub-region around Florida, and that region is consequently shown in the zoomed map reel panel. The data shown in this screenshot is yearly US county-level age-adjusted cancer mortality data from the CDC, from 1999-2020. ClusterRadar can be accessed at https://episphere.github.io/ClusterRadar.

a different perspective on the results. Each panel is interactive, allowing the user to gain more details about a specific element through mouse events such as hovering, clicking, and dragging. The panels are interactively linked: when the user interacts with one panel that interaction is reflected in the other panels. In addition to the plot panels, ClusterRadar has a tool bar with options for uploading and configuring data, enabling and disabling methods, switching coloring modes, and downloading the data. To encourage use among a diverse set of users, ClusterRadar also features a short, interactive tutorial which describes the tool's purpose and highlights its features. A video outlining the tool's key features can be found in S1 Video.

**Coloring.** The primary cluster map panel, graphical tooltip, zoomed map reel panel, and cluster assignments cell plot panel (see Fig. 1a, 1b, 1c, and 1e) share a unified coloring scheme. The tool offers multiple coloring modes: a categorical color scheme for each of the individual LISAs and an aggregate color scheme that summarizes the assignments across all enabled methods. In general, locations with similar cluster assignments (e.g., "high-high" or "hot-spot") receive the same color. The aggregate mode uses a color scale to represent the degree of agreement among different clustering methods, with more saturated colors indicating stronger agreement. Special color assignments exist for when methods seemingly contradict

one another (purple) or have a relationship that is not easily covered by other color assignment criteria (yellow). For a comprehensive explanation of the coloring schemes, including the mathematical formulas used in the aggregate mode and handling of special cases, please refer to S3 Appendix (Color assignment).

**Primary cluster map panel.** The most prominent panel in the ClusterRadar dashboard is the primary cluster map panel: a categorical choropleth plot that colors each location by its cluster assignments (see Fig. 1a). The choropleth plot is interactive: when the user hovers their cursor over a location, that location is brought into focus until they move the cursor elsewhere. The statistical density plot, cluster assignments cell plot, and statistical time-series panels are updated to show information pertaining to the in-focus location; see the individual descriptions of these panels for more details. The user can also keep a location in focus by clicking on it, meaning it will not be taken out of focus when their cursor leaves that location. When a user hovers their cursor over a location, a graphical tooltip appears showing additional information about that location (see Graphical tooltip). By default, the primary cluster map panel shows data from the most recent time point but the user can look through different time points using the time slider above the plot. Finally, when the user clicks and drags on the map, a selection box will appear allowing them to select multiple locations at once for further inspection in the zoomed cluster reel panel.

**Zoomed cluster reel panel.** The zoomed cluster reel panel (see Fig. 1c) allows the user to directly inspect how cluster assignments have changed over time in a selected sub-region of space. When the user has selected a sub-region using the primary cluster map panel, this panel will show a vertically stacked "reel" of choropleth plots—one for each time step in the dataset. The user can then scroll through this reel to see the evolution of cluster assignments over time. Each of the choropleth plots in this panel have the same interactive features as the primary choropleth, excluding the ability to select the zoomed sub-region of space.

**Density plot panel.** The density plot panel consists of density plots showing the empirical distributions of the enabled spatial statistics (see 2a and 2b), allowing the user to gain a better understanding of each statistic's distribution and significance assignments. The empirical distribution is calculated using the permutation approach described in Assessing significance. If the aggregate coloring mode is selected, then a density plot is shown for each enabled statistic, otherwise a single density plot is shown with the corresponding statistic of the selected single indicator mode. The content of the density plot panel also depends on whether or not a specific location is in focus. If so, the panel will show information regarding the local statistic(s) at that location. Otherwise, the panel will show the global statistic(s) for the whole dataset. Each density plot consists of a filled area representation of the statistic's empirical distribution, a solid red line indicating the statistic's value, and two dashed grey lines indicating the upper and lower significance boundaries at the user's chosen significance cut-off (0.05 by default). The filled area is generated using kernel-density estimation over the permuted values.

**Cluster assignments cell plot panel.** The cluster assignments cell plot panel shows a cell plot with time on the x-axis and spatial clustering method on the y-axis (see 2c and 2d). Each cell is colored according to the cluster assignment for that method at that timestep. If no location is in focus, then the cell plot shows the assignments from the global statistics. If a location is in focus, then the cell plot shows the assignments from the local statistics for that location. In the aggregate coloring mode, an additional row is added to the bottom of the cell plot which shows the aggregate color assignments. This helps the user understand how the aggregate color scheme works. The cell plot provides the most complete summary of the multi-method cluster assignments over time, but it is limited to a single location at a time.

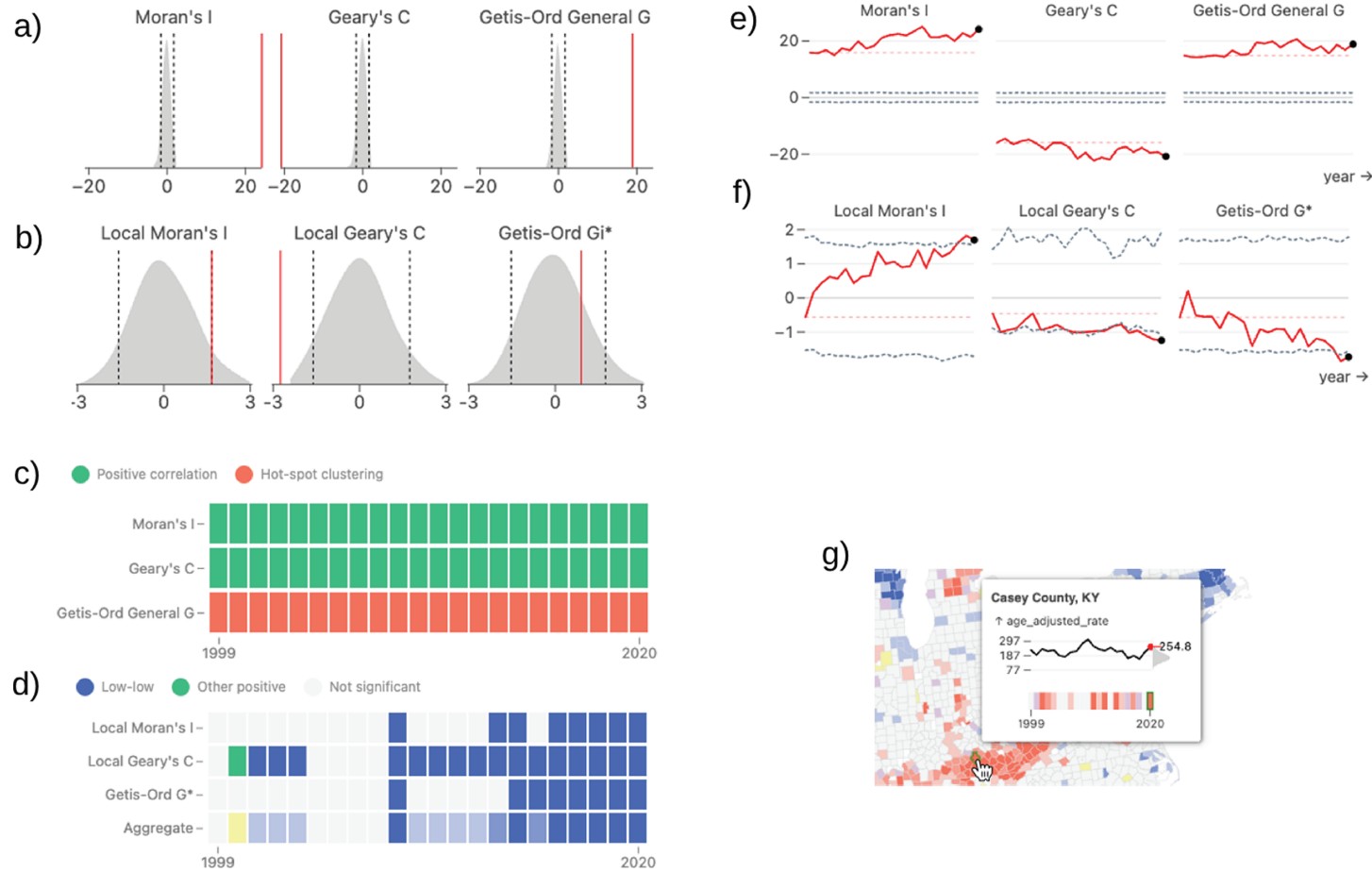

**Fig 2. The visual elements of the ClusterRadar dashboard: the density plots (a, b), cluster assignments cell plot (c, d), statistical time-series plots (e, f), and graphical tooltip (g).** The density plots depict the distribution of local (b) or global (a) indicators, with significance thresholds indicated by dashed lines. The cluster assignment cell plot illustrates global (a) or local (b) cluster assignments over time, with an additional row in (b) for the aggregate color scheme assignments. The time-series plots track local (b) or global (a) statistics over time, highlighting significance boundaries and a reference value. The graphical tooltip (g) appears when a user is hovering over a location in the map, presenting that location's name or ID, a time-series of its values, a density plot contextualizing the values, and color assignments for the chosen color mode.

**Statistical time-series panel.** The statistical time-series panel allows the user to inspect the spatial statistics over time (see Fig. 2e and 2f). If the aggregate viewing mode is selected, the time-series plot is split into sub-plots for each enabled method. Otherwise, a single plot for the current mode's method is shown. Within each plot there is a solid red line tracking the statistic's actual value over time and two dashed dark-grey lines tracking the p-value cut-offs over time. As with the cluster assignments cell plot and statistical density plot panels, the statistic(s) represented in the time-series plot depends on the current interaction state. If a specific location is in focus, then the plot will show the enabled local statistic(s) for that location. If no location is in focus, then the plot will show the enabled global statistic(s) for the entire dataset.

**Graphical tooltip.** The graphical tooltip appears on the choropleth plots when the user hovers over a specific location, in either the main cluster map panel or the zoomed cluster reel

panel (see Fig. 2g). The tooltip allows the user to quickly get more information about a location. At the top of the tooltip is location's name or ID. Below that is a time-series plot showing the location's value over time. The axis is marked with the mean (over the whole dataset across all time points) and the mean plus or minus 3 standard deviations. Attached to the right axis of the time-series is a density plot showing the distribution of values across all time points. On this density plot, the current value is shown as red line labelled with the value in text. At the bottom is a single row cell plot showing the color assignments over time for the in-focus location; the exact coloring shown depends on the enabled mode and is identical to the coloring shown on the map.

**Implementation details.** ClusterRadar is implemented in vanilla JavaScript and runs entirely on the client-side in the user's web browser. The basic graphical elements of the dashboard were rendered using the Observable Plot library, with interaction and additional visual elements added using D3, HTML, and CSS. Currently, there is no JavaScript library which supports the full set of spatial indicators required in ClusterRadar. The jsgeoda library supports most of them using WebAssembly but it doesn't provide estimates of the empirical distributions or significance cut-offs. Therefore, all methods were re-implemented in JavaScript. Calculating the LISA results is particularly computationally intensive due to the permutation-based significance testing, which requires re-calculating the statistic many times for each location. To ensure a responsive user interface and parallelize computation, web workers are used. When results for a given configuration are calculated, they are cached on the user's machine using IndexedDB so that the user does not need to re-run the calculations every time they visit the tool. We evaluated the scalability of the tool (see Fig. 3), and found that it is practical for reasonably large datasets (such as US county-level data, with around 3,200 areas), but would likely face memory issues for US census level data (with around 8.2 million areas). For more in-depth technical evaluations, including performance comparisons with jsgeoda, see S4 Appendix (Performance) and S5 Appendix (Memory usage).

**Fig 3. Basic technical evaluation of ClusterRadar showing memory burden and run times on a Macbook with an M1 Pro chip and 16gb of RAM.** Performance and memory burden grow linearly with the number of areas. Per-process memory limits vary considerably, but an assumed memory limit of 4GB puts the estimated limit on number of features at around 1.5 million. This is sufficient for many use cases, but may be insufficient for fine-grained geographic units such as US census blocks (the entire US is comprised of over 8 million census blocks).

## Usage scenario: US cancer mortality

**Insights.** To show how ClusterRadar may be applied to real world data, we analyzed age-adjusted US county-level cancer mortality with a domain expert in geospatial epidemiology. The data was collected from CDC Wonder, with a filter for the "malignant neoplasms (C00-C97)" group of the ICD-10 113 cause list. The data is yearly from 1999 to 2020. We uploaded the data to the ClusterRadar tool and inspected the results in the dashboard. We noticed the following:

- **I1**: An overall trend of increased global spatial autocorrelation, especially according to the Moran's I statistic (see Fig. 4a). This is immediately apparent from the statistical time-series panel. The expert suggested this increase in spatial autocorrelation may be a result of overall increased spatial structure in cancer risk factors, including growing spatial disparities in smoking, obesity, and socioeconomic factors. The interactivity of the primary cluster map panel allows the analyst to find specific areas of the US where the increase in spatial structure is most apparent (see I2 for an example).

- **I2**: An emerging cold-spot in the north-eastern US around New Jersey, Pennsylvania, New York, and Connecticut (see Fig. 4d). The tooltip shows a steady decline in age-adjusted mortality rates since 1999, with a concomitant increase in assignment to low clusters (see Fig. 4e). The cluster assignments cell plot shows that these clusters began to emerge in the early 2000s, and that the Geary's C method was generally the earliest to detect significant clustering. The statistical time-series plots reflect the gradual emergence of these clusters, with increasing local spatial autocorrelation from Local Moran's I and Geary's C, and cold-spot clustering from Getis-Ord Gi* (see Fig. 4f). These plots show that the strength of these cluster assignments has been steadily growing over time, a result that would have been missed in a static analysis. The expert suggested that this may be caused by growing affluence in these coastal and near-coastal counties, factors associated with lower cancer mortality. Shifting demographics in these counties may be a contributing factor, though further investigation would be required to disentangle all the possible factors involved.

- **I3**: A large and fluctuating hot-spot in the south, around Kansas, Tennessee, and Ohio. The expert noted that her eye was drawn immediately to this hot-spot, and said that the higher rates of cancer in this region of the US are well-studied, believed to be driven by a variety of risk factors including poverty, smoking rates, and obesity. The fluctuating nature of the hot-spot becomes apparent when interacting with the time slider, or by hovering over the hot-spot's counties and inspecting any of the temporal plots (the cell plot, the time-series plot, the zoomed cluster reel panel, or the tooltip). The expert agreed that the fluctuating nature of this hot-spot is a good example of why longitudinal analysis is important: any one time-point may present a misleading picture of confidence in the cluster's exact shape, but overall we can observe a consistent cluster of high cancer rates in that general region of the US.

- **I4**: A large cold-spot encompassing much of the western US. The expert noted that the cold spots in the coastal counties are expected, given the typically higher socioeconomic status of those counties. She also cautioned about placing too much emphasis on the other counties due to low populations—the cluster draws the eye due to its sheer geographic size, but encapsulates a relatively small number of people. On top of that, there is a lot of suppressed data in that part of the country due to small population sizes.

- **I5**: A sudden hot-spot in northern Michigan in 2020 (see Fig. 4b). The graphical tooltip shows that this primarily results from a sudden spike in Oscoda County, MI (see Fig. 4c).

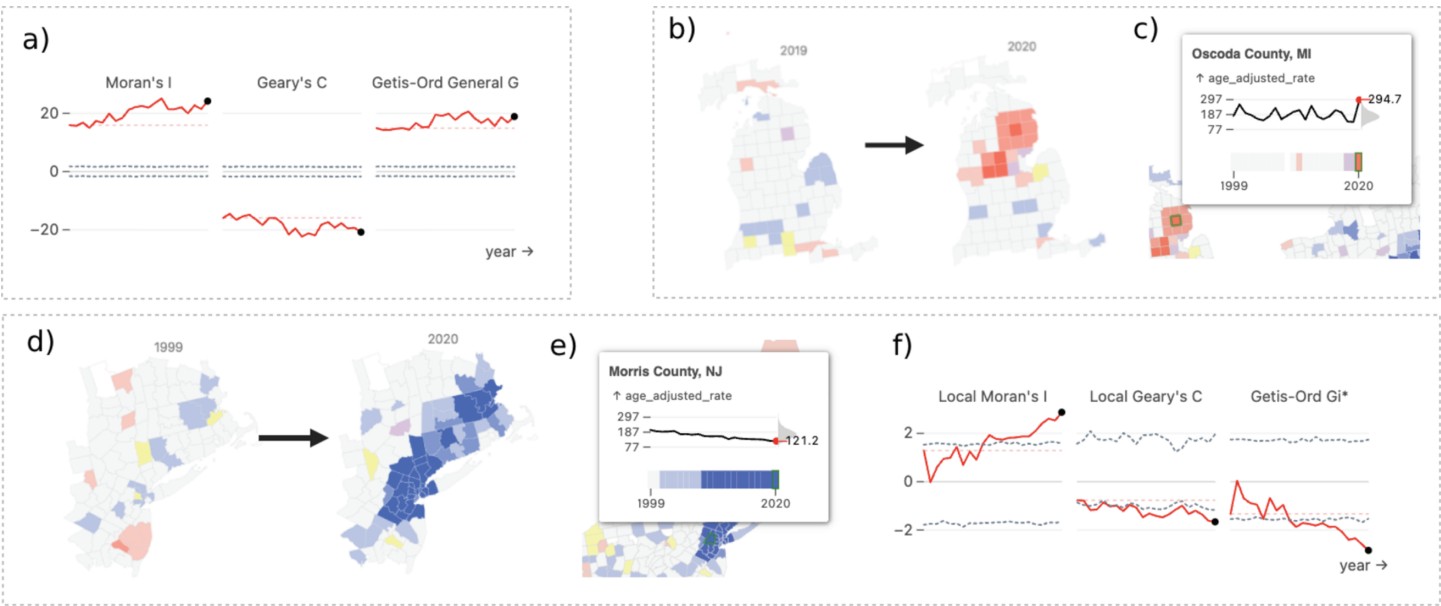

**Fig 4. Example snapshots of ClusterRadar on a dataset of US county-level age-adjusted mortality rates, from the CDC.** These illustrate some of the potential insights that ClusterRadar can help uncover. (a) A snapshot of the statistical time-series panel showing the global spatial statistics. There appears to be a general increase in these statistics, especially for Moran's I. (b) An edited snapshot of the cluster reel panel, showing the sudden appearance of a high cluster in Northern Michigan in 2020. (c) A snapshot of the primary cluster map panel in which the user is hovering over a central area from the high cluster in Northern Michigan. The time-series shows a sudden spike in the age-adjusted mortality rate in 2020 and the cell plot shows a concomitant assignment to a high cluster. (d) An edited snapshot of the cluster reel panel, showing the emergence of a low cluster in the east coast around New Jersey, New York, and Connecticut. (e) A snapshot of the primary cluster map panel in which the user is hovering over a central area (Morris County, NJ) from the low cluster on the east coast. The time-series show a gradual decline in age-adjusted mortality rates in that county. (f) A snapshot of the statistical time-series panel showing the local statistics for Morris County, NJ.

Oscoda County has a relatively small population which can lead to unstable rates. This suggests that this particular cluster may just be noise, a result which emphasizes the importance of longitudinal analysis of spatial clusters.

The expert emphasized that these observations are purely exploratory, and that robust statistical testing would need to be done to investigate them further. It is worth noting that many of these insights (I1, I2, I3, and I5) would not have been apparent from a static analysis, further validating the longitudinal approach to spatial analysis facilitated by ClusterRadar. The tool's varied graphical plots and robust interactive features convey the temporal structure of both the data and the results, exposing insights that would not have naturally emerged from existing static tools.

**Quantitative evaluation.** To provide some additional evaluation of ClusterRadar, we have performed an inter-method and temporal evaluation of the cluster assignments on the previously described cancer mortality dataset. The results of the inter-method evaluation can be found in Fig 5 and the results of the temporal evaluation can be found in Fig 6. The inter-method evaluation shows a substantial degree of disagreement between the methods when at least one method's assignment is statistically significant. For instance, only 32.1% of dual-method comparisons involving a 'high-high' label were in agreement. Most contradictions arose because one method detected statistically significant spatial structure and the other method did not. It is worth noting that some apparent contradictions occur due to the differing nature of the methods, rather than a direct disagreement about the spatial structure at the focal location. For example, you may notice the substantial number of 'high-high' results

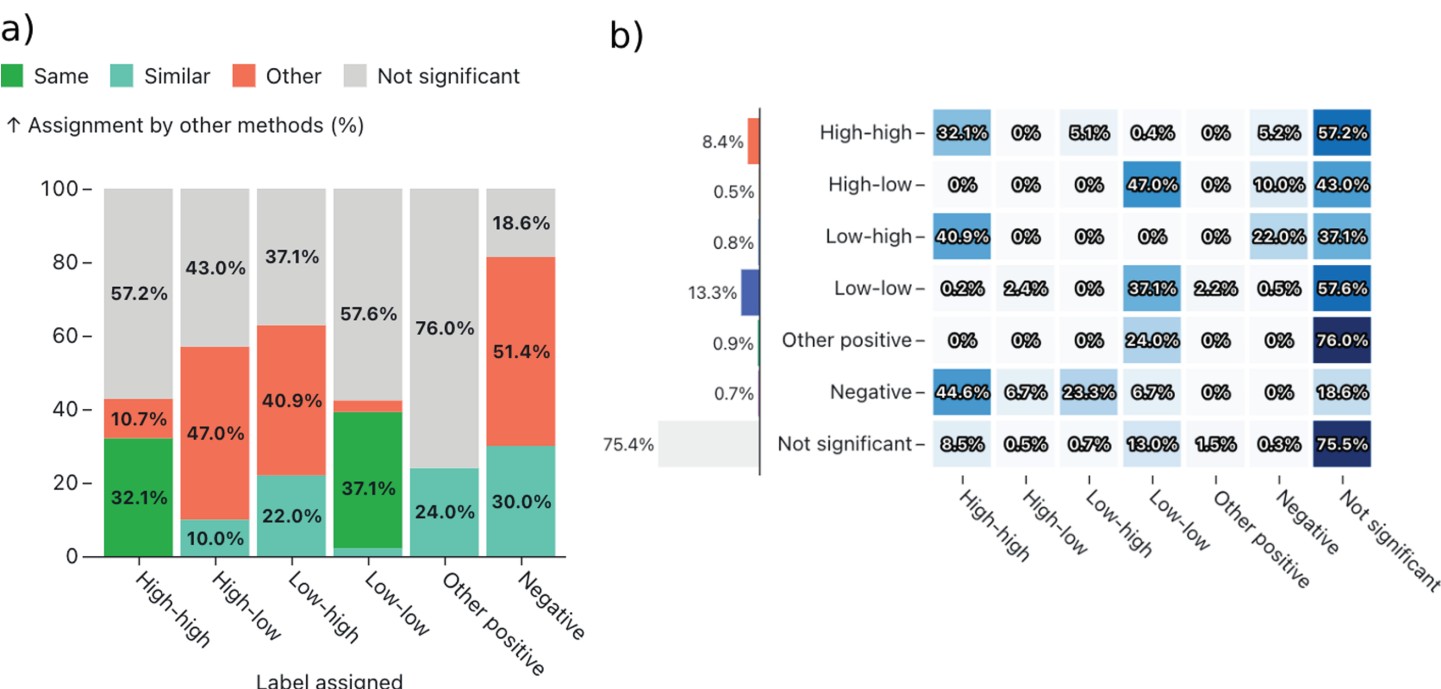

**Fig 5. A quantitative evaluation of the multi-method spatial clustering results generated by ClusterRadar, on a US county-level cancer mortality dataset.** (a) A stacked bar chart breaking down the nature of between-method assignment comparisons, by label. e.g. the leftmost bar shows that 32.1% of between method comparisons for the 'high-high' label were comparisons where both assignments were 'high-high', 57.2% were comparisons with a 'not significant' label, and 8.1% were comparisons with contrasting labels. The "similar" category is used when methods are not necessarily in agreement but don't contradict each other (e.g. 'high-high' and 'other positive spatial autocorrelation.') (b) A cell plot providing a more detailed breakdown of between-method assignment comparisons e.g. the third cell on the top row shows that 5.1% of comparisons involving a 'high-high' label are comparisons where the other label is 'low-high'. The bar chart on the left shows the total number of assignments for the corresponding label.

which were assigned a label indicating negative spatial autocorrelation (i.e. 'low-high' or 'negative') by another method. A closer inspection of the results reveals that this occurs because of Getis-Ord Gi* which, unlike the other methods, doesn't detect spatial autocorrelation but instead directly detects hot and cold spots. Consequently, a location with a relatively low value may still be detected as a 'hot-spot' (equivalent to 'high-high' in ClusterRadar) by Getis-Ord Gi*, whereas this would likely be assigned 'low-high' by Local Moran's I and 'negative' by Local Geary's C. Situations like this expose some of the difficulties associated with directly comparing spatial clustering results across methods. However, it also shows the value of a multi-method approach, as different methods expose different aspects of spatial structure, leading to a more complete understanding of the data.

The temporal evaluation of the cancer mortality dataset shows the dynamic nature of the results, and furthers the argument for longitudinal investigation as a important step in spatial analysis. As Fig. 6a shows, even a single time-step difference has a substantial effect on the results, with many locations exhibiting a different result at an immediately prior time-step. If a longitudinal investigation was not performed, one of these results would be the only information provided to the analyst and crucial dynamic information about the detected spatial clusters (such as their stability) would be missed. Fig. 6b shows the variability in how different methods respond to the data's dynamics. For instance, only 1.9% of locations assigned a 'high-high' result by Local Moran's I were assigned a 'low-low' result by Local Moran's I at any other timestep, but for Local Geary's C this percentage is much higher at 32.5%. In other

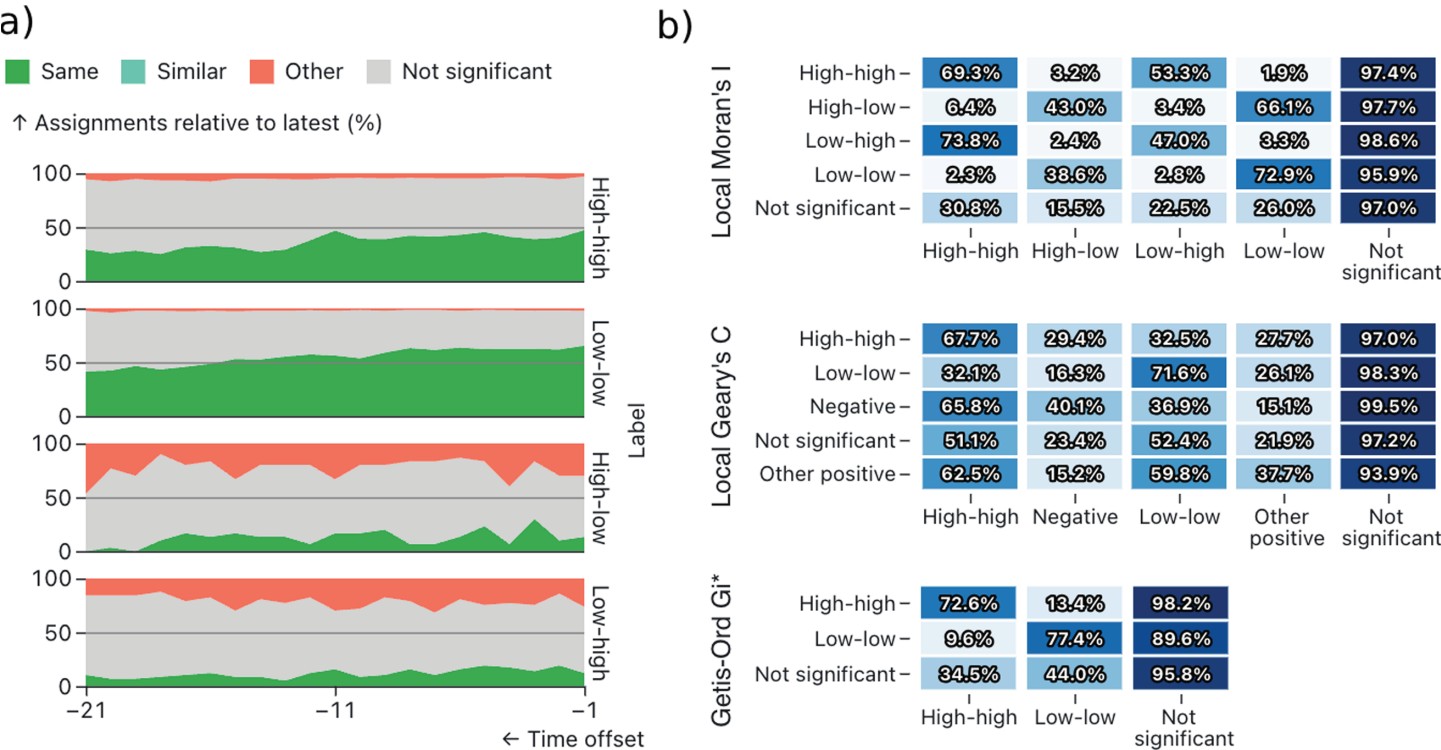

**Fig 6. A quantitative evaluation of the longitudinal spatial clustering results generated by ClusterRadar, on a US county-level cancer mortality dataset.** (a) An area chart showing the percentage of agreement between the label assigned at the most recent time-point for a location, and the labels assigned at previous time-points. e.g. The topmost area chart shows that a location assigned a 'high-high' label at the most recent time-point was also assigned 'high-high' at the previous time-point in around 50% of instances, and was assigned 'High-high' at the earliest time-point in around 29% of instances. (b) Percentage of labels shared by a single location's time-series, by method. e.g. the third cell in the top row of the topmost cell plot shows that 53.3% of locations that were assigned a 'high-high' label by the Local Moran's I method at one time-point were assigned a 'low-high' label by the same method at some other time-point in the past or future.

words, the dynamics of the results vary considerably by the chosen method, an observation which supports the need for a simultaneous multi-method and longitudinal analysis.

Together, the quantitative evaluations discussed here demonstrate the additional insight which can be naturally obtained in ClusterRadar which would be time consuming or impossible in other tools. The temporal and methodological variability present in the real-world dataset indicate that a longitudinal, multi-method approach is necessary to fully contextualize and understand the spatial clusters present in the dataset.

## Initial feedback

To provide an initial evaluation of ClusterRadar, we sent an evaluation survey to individuals with a diverse set of analytical interests. The NIH IRB determined that this survey is not human subjects research, and therefore does not require IRB approval. Consent for all participants was obtained informally by email. All seven survey participants have a research background, with interests spanning data science, public health, geospatial epidemiology, and computer science. The form and a snapshot of its responses at the time of writing are available in S6 File. All participants responded that they had a basic understanding of spatial data and analysis, but only one participant claimed expert knowledge of the methods; the remaining participants expressed a mixture of unfamiliarity and basic understanding. This is useful to test ClusterRadar's suitability for non-expert users (D4).

When asked if the insight provided by temporal analysis of spatial clusters is worth the additional complexity, 5 of 7 (71.4%) of participants answered affirmatively, and two participants answered that it is potentially useful but may not be worth the additional complexity. When asked whether the comparison of multiple clustering methods is useful, 3 of 7 (42.9%) of users answered affirmatively, 3 of 7 (42.9%) answered that it is potentially useful but may not be worth the additional complexity, and 1 of 7 answered they remained uncertain but enjoyed viewing the comparison across methods.

In the evaluation of the tool itself, 4 of 7 (57.1%) participants said ClusterRadar is successful in its primary goal of making the analysis of spatial clusters over time more accessible. The remaining participants said it is "somewhat" successful, with one stating that the tool is more descriptive than analytical. One participant expressed a desire for more information on the methods involved in the tool and how they differ.

The participants were asked the extent to which the major features of ClusterRadar are useful. The primary cluster map panel was deemed "very useful" by all 7 participants. The following features received a mixture of "very useful" and "somewhat useful" responses: the graphical tooltip (85.7% "very useful") the zoomed map reel panel (71.4% "very useful"), the statistical time-series panel (71.4% "very useful"), the cluster assignments cell plot panel (42.9% "very useful"), the aggregate color scheme (71.4% "very useful"). The density plot panel received 57.1% "very useful", 28.6% "somewhat useful", and one response of "not useful": that participant commented that they did not know how to interpret the density plots. All participants said that the tool's implementation as a web-application was "very useful". One participant commented that the plot choices were "very insightful".

To evaluate ClusterRadar's potential utility as a research tool, and to improve its design, we asked seven fellow researchers to provide anonymous feedback on the tool's methodology and user experience. The participating researchers spanned a variety of backgrounds, including data science, public health, and geospatial epidemiology. All participants responded that they had a basic understanding of spatial data and analysis, but only one participant claimed expert knowledge of the specific methods applied; the remaining participants expressed a mixture of unfamiliarity and basic understanding. This is useful to test ClusterRadar's suitability for non-expert users (D4). The majority of responses confirmed that the tool is successful in making the longitudinal analysis of spatial clusters more accessible to researchers, though some expressed concerns that the complexity of the methods employed in ClusterRadar may be excessive for their research.

## Discussion

In this paper, we have introduced ClusterRadar, a web-tool which confronts the challenge of multi-method, temporal exploration of spatial clusters using multi-faceted interactive visualization. In this section, we will discuss some of the problems we tackled while creating this tool.

A core challenge in multi-method, spatiotemporal cluster analysis is managing the potentially large volume of results. In our example (see Usage scenario: US cancer mortality), applying just 3 clustering methods across 22 time points to 3,143 counties yields over 200,000 individual results. ClusterRadar addresses this complexity through a multi-plot dashboard that follows the visual analytics principle of "overview first, details-on-demand" (D3). This approach allows users to gain a broad understanding of the results before drilling down into specific details. This philosophy is reflected in several parts of the tool, including the responsive nature of the density, cell, and time-series panels. Each of these visualizations initially represent global statistics but changes to represent local statistics when the user interacts with

a specific location. The global statistics provide the "overview", and the local statistics provide the "details-on-demand". This approach is well supported by the visual analytics literature, and ClusterRadar follows established principles to simplify the difficulties which may arise when exploring multiple visualizations at once, including the principle of maintaining the user's mental map between plots using linked interaction [80]. Initial feedback from a users regarding the plots was generally positive. However, some plots (e.g. the primary cluster map) received more positive feedback than others (e.g. the statistical density plots), so a future effort to provide alternatives and collect feedback may benefit the tool's overall success in representing complex results.

The results generated by ClusterRadar are both spatial and multi-variate/multi-method (D2). Visualizing the spatial distribution of multi-variate results is a well-documented challenge in the geospatial visualization literature and a broad range of solutions have been proposed. Once again, we took a visual analytics approach, with the aggregate color scheme offering the overview, and user interaction providing further details. The aggregate color scheme provides a way for the user to inspect the level of agreement between results at a glance. While some finer distinctions between methods might be obscured in this simplification, it provides a valuable starting point. For users wishing to delve deeper into the nuances, the interactive, linked views provide the necessary granularity to dissect and compare results with greater precision.

A challenging element of the ClusterRadar results is that they are both spatial and temporal (D1)—elements that can be difficult to visualize together. Like with multi-variate spatial data, the problem of representing spatiotemporal data in plots is well documented in the visualization literature. There are two key approaches: contrasting time and contrasting space [59]. In the former, some representation of the temporal results (e.g. a time-series) is plotted at each geospatial location, and in the latter separate geospatial plots are used to represent data from different time-points. The contrasting time approach can be overwhelming when there are a lot of geospatial locations and the temporal results are complex, as is the case in ClusterRadar, and so we decided on the contrasting space approach. This was achieved using the cluster map reel panel, which stacks choropleths from different time points on top of each other, and the interactive time-slider in the primary cluster map panel, which allows users to "animate" the cluster results by viewing data from different time-points. The statistical time-series panel, cluster assignments cell plot panel, and graphical tooltip all provide additional information on temporal elements of the results.

A key element in the success of novel applications is their usability and accessibility to the target user base. In this case, our target user base is anybody interested in the analysis of spatial clusters. While a working knowledge of spatial data, statistical principles, and common visualizations is beneficial, the ClusterRadar tool is designed to minimize the required expertise. It does so in a few ways: applying a varied set of visualizations to streamline the interpretation of the results (D3), employing a goal-focused design philosophy (D4), and distributing the application on the web (D5). Importantly, the user can access the application from any modern browser without the need for installation. ClusterRadar eliminates the need for specialized technical knowledge of the methods; users simply upload data, configure settings, and receive results. The initial feedback from users suggests the success of this approach, though additional work is required to address certain confusing elements. Several individuals expressed confusion over aspects of the tool which could potentially be cleared up through better documentation and a refined tutorial.

When inspecting the cancer mortality data in Usage scenario: US cancer mortality, the expert analyst expressed the opinion that ClusterRadar is primarily an exploratory tool, rather than an analytical one. The distinction between these concepts is important: exploratory tools

can uncover patterns worthy of further analysis (hypothesis generation), but are not necessarily appropriate for more rigorous inspection (hypothesis testing). Exploratory data analysis is becoming common across disciplines, including those with well-established analytical traditions such as epidemiology, driving a need for exploratory tools such as ClusterRadar [86,87]. ClusterRadar allows the user to download the results, which an experienced analyst could then inspect in the analytical environment of their choice (e.g. R). Further work could improve the analytical appeal of ClusterRadar by supporting p-value correction or other techniques important in confirmatory spatial cluster analysis.

## Conclusion

ClusterRadar advances the analysis of complex spatial datasets by offering a user-friendly environment specifically designed for interpreting temporal and multi-method spatial clustering results. By combining diverse visual elements with linked interaction, ClusterRadar tackles the challenges of representing multi-variate, spatiotemporal data. The tool's goal-driven design democratize spatial cluster analysis, empowering users across disciplines to uncover spatial patterns within their data. ClusterRadar runs entirely in the user's browser and does not require a server or installation, preserving the privacy of the user's data and ensuring the tool's longevity.

## Supporting information

**S1 Video.** Video outlining the main features of the ClusterRadar tool.
(MP4)

**S2 Appendix (Assessing significance).** Greater detail on how permutation tests are used within ClusterRadar to assess significance.
(PDF)

**S3 Appendix (Color assignment).** Greater detail on how colors are assigned in ClusterRadar.
(PDF)

**S4 Appendix (Performance).** Performance benchmarks for ClusterRadar.
(PDF)

**S5 Appendix (Memory usage).** Memory usage benchmarks for ClusterRadar.
(PDF)

**S6 File.** User survey and responses.
(PDF)

## Author contributions

**Conceptualization:** Lee Mason, Jonas S. Almeida.

**Data curation:** Lee Mason.

**Formal analysis:** Lee Mason.

**Investigation:** Lee Mason.

**Methodology:** Lee Mason.

**Resources:** Lee Mason.

**Software:** Lee Mason.

**Supervision:** Blánaid Hicks, Jonas S. Almeida.

**Validation:** Lee Mason, Jonas S. Almeida.

**Visualization:** Lee Mason.

**Writing – original draft:** Lee Mason.

**Writing – review & editing:** Lee Mason, Blánaid Hicks, Jonas S. Almeida.

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
