## [Decision Letter · Decision Letter 0]

15 Jan 2025

PONE-D-24-24211ClusterRadar: an interactive web-tool for the multi-method exploration of spatial clusters over timePLOS ONE

Dear Dr. Mason,

Thank you for submitting your manuscript to PLOS ONE. After careful consideration, we feel that it has merit but does not fully meet PLOS ONE’s publication criteria as it currently stands. Therefore, we invite you to submit a revised version of the manuscript that addresses the points raised during the review process.

We look forward to receiving your revised manuscript.

Kind regards,

Nazarudin Safian, PhD

Academic Editor

PLOS ONE

Journal Requirements:

4. You indicated that ethical approval was not necessary for your study. We understand that the framework for ethical oversight requirements for studies of this type may differ depending on the setting and we would appreciate some further clarification regarding your research. Could you please provide further details on why your study is exempt from the need for approval and confirmation from your institutional review board or research ethics committee (e.g., in the form of a letter or email correspondence) that ethics review was not necessary for this study? Please include a copy of the correspondence as an ""Other"" file.

6. Thank you for stating the following financial disclosure:

“Research reported in this publication was supported by the National Cancer Institute of the National Institutes of Health (CAS 10901)”

7. We note that [Figures 1,3 and 4] in your submission contain [map/satellite] images which may be copyrighted. All PLOS content is published under the Creative Commons Attribution License (CC BY 4.0), which means that the manuscript, images, and Supporting Information files will be freely available online, and any third party is permitted to access, download, copy, distribute, and use these materials in any way, even commercially, with proper attribution. For these reasons, we cannot publish previously copyrighted maps or satellite images created using proprietary data, such as Google software (Google Maps, Street View, and Earth). For more information, see our copyright guidelines: http://journals.plos.org/plosone/s/licenses-and-copyright.

a. You may seek permission from the original copyright holder of Figures 1,3 and 4 to publish the content specifically under the CC BY 4.0 license.  

Additional Editor Comments:

Overall Impression:

The manuscript introduces a web tool, ClusterRadar, for exploring spatiotemporal clusters, addressing a genuine need. The authors have clearly articulated their design considerations and developed a user-friendly interface. However, several areas of the technical details, depth of analysis, and robustness could be improved.

i. Structure and Organization:

(Page 2, Line 42): The transition from the introduction to the background is abrupt. Include a brief introductory sentence to set up the background section, like "To understand the value of this tool, the following background information is necessary."

(Page 3-4, Lines 110-162): The "Related work" section is lengthy and could be more concise. Summarize the main points to more effectively highlight the authors' unique contribution and improve the flow of the document.

** (Page 6-8, Lines 233-326):** The "Methods" section is very long and may lose the reader's attention. Consider adding subheadings to better organize the content and improve readability.

(Page 9, Lines 358-368): The results section focuses excessively on the tool’s description and not enough on the derived insights. Reframe the section to focus on the analytical outputs and their potential applications.

(Page 14, Lines 516-555): The use case scenario could be more robust, fully demonstrating all aspects of the tools interactivity and visualization abilities. Expand this section to include more in-depth analysis of the datasets.

ii. Technical Content:

(Page 6, Line 240-242): The justification for using a binary weight matrix could be stronger. Discuss the alternatives and their relative advantages and disadvantages.

Throughout the document: The manuscript is long and does not emphasize the more technical aspects of the tool, such as implementation details, limitations, scalability, and computational efficiency. Highlight these points and provide supplementary documents if necessary.

Throughout the document: There is not much discussion of the technical limitations, and where the tool could potentially fail. Address the limitations of this tool, both technical and methodological, to establish its intended use cases.

iii. Scientific Merit:

(Page 14, Lines 516-555): The analysis presented in the "Usage scenario" section is brief and the conclusions drawn are somewhat basic. Explore more of the tool’s functionality and provide a more in-depth analysis of the cancer mortality dataset, with stronger conclusions.

The tool lacks robust validation. Include a section to validate the tool by comparing its performance with existing tools on standardized datasets.

The manuscript does not include a deep discussion of the caveats in the analysis, particularly when combining multiple methods. Discuss the theoretical and practical caveats of combining different clustering methods.

iv. Overall Suitability for Publication:

The manuscript is rather lengthy and contains information that may distract from the core contributions of the paper. Revise the document to emphasize the core contributions, and remove or summarize some of the background information and technical details.

The lack of robustness in the analysis, the lack of validation of the tool, and the lack of addressing methodological caveats could hamper its suitability for publication. Address these issues and create a supplementary document to include important but distracting technical details.

Summary of Key Points

The manuscript introduces a functional tool with great potential, but needs more emphasis on analytical rigor and robustness, with more efficient presentation. The above suggestions, including more concrete analyses, will help in the presentation of the core contributions.

Reviewers' comments:

Reviewer's Responses to Questions

**Comments to the Author**

1. Is the manuscript technically sound, and do the data support the conclusions?

Reviewer #1: Yes

2. Has the statistical analysis been performed appropriately and rigorously? 

Reviewer #1: Yes

3. Have the authors made all data underlying the findings in their manuscript fully available?

Reviewer #1: Yes

4. Is the manuscript presented in an intelligible fashion and written in standard English?

Reviewer #1: Yes

5. Review Comments to the Author

Reviewer #1: The Cluster Radar is great, however, I would like to about how will we handle the missing data ?

The US cancer data from CDC that you showed in the video is excellent that you can capture and found ONE county in New Jersey. However, I would like to know that if I cannot find the complete data set like that. How do we are going to deal with the missing data.

6. PLOS authors have the option to publish the peer review history of their article (what does this mean?). If published, this will include your full peer review and any attached files.

Reviewer #1: **Yes: **Uthumporn Panitanarak

---

## [Author Response · Author response to Decision Letter 1]

4 Mar 2025

Dear Dr Safian,

We would like to thank you and the reviewer for the time and effort you have dedicated to considering our manuscript and providing helpful feedback. We are delighted to have the opportunity to revise and resubmit our work. We have carefully considered the comments and suggestions and have made changes to the manuscript accordingly.

Reviewer Comments

1. “The transition from the introduction to the background is abrupt. Include a brief introductory sentence to set up the background section, like "To understand the value of this tool, the following background information is necessary."”

We have added an additional transitionary sentence to the end of the Introduction section.

2. “The "Related work" section is lengthy and could be more concise. Summarize the main points to more effectively highlight the authors' unique contribution and improve the flow of the document.”

We have re-written each sub-section in ‘Related work’ to be shorter and more explicitly focused on the unique contributions of our work. The one exception is the ‘Spatial clustering software’ sub-section, which we have kept the same because it covers the related work most relevant to our proposed tool.

3. “The "Methods" section is very long and may lose the reader's attention. Consider adding subheadings to better organize the content and improve readability.”

We have re-organized this section with an additional level of sub-headings, and we have substantially shortened the ‘Assessing significance’ sub-section, moving the more specific details to an appendix.

4. “The results section focuses excessively on the tool’s description and not enough on the derived insights. Reframe the section to focus on the analytical outputs and their potential applications.”

We have substantially reduced the size of the ‘Coloring’ subsection, moving the detail to an appendix. We have also added in technical evaluation, expanded discussion of the survey results, provided stronger commentary in the usage scenario, and added a quantitative evaluation. We believe this places greater emphasis on the analytical results and the unique contributions of the tool.

5. “The use case scenario could be more robust, fully demonstrating all aspects of the tools interactivity and visualization abilities. Expand this section to include more in-depth analysis of the datasets.”

“The analysis presented in the "Usage scenario" section is brief and the conclusions drawn are somewhat basic. Explore more of the tool’s functionality and provide a more in-depth analysis of the cancer mortality dataset, with stronger conclusions.

We have added additional detail to the usage scenario insights discussion, and more clearly linked the insights back to specific features of the tool. We have added a more in-depth analysis of the results on the cancer mortality dataset in a new ‘Quantitative evaluation’ subsection.

6. “The justification for using a binary weight matrix could be stronger. Discuss the alternatives and their relative advantages and disadvantages.”

We have added a sentence justifying the binary weight matrix.

7. “Throughout the document: The manuscript is long and does not emphasize the more technical aspects of the tool, such as implementation details, limitations, scalability, and computational efficiency. Highlight these points and provide supplementary documents if necessary.

Throughout the document: There is not much discussion of the technical limitations, and where the tool could potentially fail. Address the limitations of this tool, both technical and methodological, to establish its intended use cases.”

We have added additional text and a new figure to the ‘Implementation details’ section, detailing the scalability of the tool and the feasibility of using it on real-world geospatial datasets. We have provided more detailed technical evaluation in two new appendices: S4 and S5.

8. “The tool lacks robust validation. Include a section to validate the tool by comparing its performance with existing tools on standardized datasets.”

We have added an appendix (S4 Appendix) which compares ClusterRadar’s performance to a web library (jsgeoda) that provides spatial clustering methods. We chose to exclude this comparison from the main body because no existing software or library, on the web or otherwise, provides ClusterRadar’s functionality in full. ClusterRadar’s unique contribution is it’s uniquely streamlined approach to multi-method and longitudinal analysis. We agree this comparison is important but we felt placing it in the main text might mislead the user into thinking ClusterRadar is a direct alternative to or replacement for other software, when it is instead a new concept which synthesizes existing methodologies in a unique way.

9. “The manuscript does not include a deep discussion of the caveats in the analysis, particularly when combining multiple methods. Discuss the theoretical and practical caveats of combining different clustering methods.”

The new ‘Quantitative evaluation’ section was added to add more depth and concrete detail to the analysis.

10. “The Cluster Radar is great, however, I would like to about how will we handle the missing data ?”

Both the ClusterRadar processing pipeline and the dashboard are equipped to handle missing data .The example cancer mortality data displayed by default on the dashboard contains a substantial degree of missing data due to data suppression (e.g in Colorado, Nebreska, etc.). The best way to see how the dashboard handles missing data is to hover over the uncolored counties.

We’d like to thank Dr. Panitanarak for their detailed, thorough, and specific suggestions for our manuscript. We believe all the changes suggested are exceptionally valuable contributions to the paper and we are very grateful for their input.

Editor Comments

1. “Please provide additional details regarding participant consent […] If the need for consent was waived by the ethics committee, please include this information.”

“Could you please provide further details on why your study is exempt from the need for approval and confirmation from your institutional review board or research ethics committee (e.g., in the form of a letter or email correspondence) that ethics review was not necessary for this study?”

We have obtained and attached a declaration of “Not Research” for the survey performed in ClusterRadar. This is a waiver category at the NIH indicating that IRB review and approval is not required. This is because the survey is not counted as generalizable research as it only seeks to validate the basic operation and utility of the tool rather than make generalizable statements about human behavior or a proposed scientific idea.

2. “We note that the grant information you provided in the ‘Funding Information’ and ‘Financial Disclosure’ sections do not match.”

“Please state what role the funders took in the study. If the funders had no role, please state … Please include this amended Role of Funder statement in your cover letter”

This project received no specific funding as it was conducted as part of the NIH’s intramural program. As requested, I have added a line to the cover letter explaining that. NIH guidelines require us to mention the CAS number in the paper, but this number is not the same as a grant number and isn’t the same as receiving specific funding from an external source.

3. “We note that [Figures 1,3 and 4] in your submission contain [map/satellite] images which may be copyrighted.”

No copyrighted map or satellite images were used in the figures. The maps present in the figures were programmatically generated by us using the open-source Observable Plot graphics library. The geographic outlines used were obtained from the Census Bureau and are public domain.

Thanks to Dr Safian, and again to Dr Panitanarak, for considering our manuscript. We are glad to have to chance to resubmit our work with the suggested improvements, and we look forward to hearing your response.

Sincerely,

Lee Mason

Division of Cancer Epidemiology and Gene8cs, Na8onal Cancer Ins8tute 9609 Medical Center

Drive, Rockville, Maryland 20850

Email: masonlk@nih.gov

---

## [Editor Report · Decision Letter 1]

21 Mar 2025

ClusterRadar: an interactive web-tool for the multi-method exploration of spatial clusters over time

PONE-D-24-24211R1

Dear Dr. Mason,

We’re pleased to inform you that your manuscript has been judged scientifically suitable for publication and will be formally accepted for publication once it meets all outstanding technical requirements.

Kind regards,

Nazarudin Safian, PhD

Academic Editor

PLOS ONE
---

## [Editor Report · Acceptance letter]

PONE-D-24-24211R1

PLOS ONE

Dear Dr. Mason,

I'm pleased to inform you that your manuscript has been deemed suitable for publication in PLOS ONE. Congratulations! Your manuscript is now being handed over to our production team.

Kind regards,

on behalf of

Professor Nazarudin Safian

Academic Editor

PLOS ONE